# The *mr*-MDA: An Invariant to Shifting, Scaling, and Rotating Variance for 3D Object Recognition Using Diffractive Deep Neural Network

**DOI:** 10.3390/s22207754

**Published:** 2022-10-12

**Authors:** Liang Zhou, Jiashuo Shi, Xinyu Zhang

**Affiliations:** 1National Key Laboratory of Science & Technology on Multispectral Information Processing, Huazhong University of Science & Technology, Wuhan 430074, China; 2School of Artificial Intelligence & Automation, Huazhong University of Science & Technology, Wuhan 430074, China

**Keywords:** diffractive neural network, deep learning, computer vision

## Abstract

The diffractive deep neural network (D^2^NN) can efficiently accomplish 2D object recognition based on rapid optical manipulation. Moreover, the multiple-view D^2^NN array (MDA) possesses the obvious advantage of being able to effectively achieve 3D object classification. At present, 3D target recognition should be performed in a high-speed and dynamic way. It should be invariant to the typical shifting, scaling, and rotating variance of targets in relatively complicated circumstances, which remains a shortcoming of optical neural network architectures. In order to efficiently recognize 3D targets based on the developed D^2^NN, a more robust MDA (*mr*-MDA) is proposed in this paper. Through utilizing a new training strategy to tackle several random disturbances introduced into the optical neural network system, a trained *mr*-MDA model constructed by us was numerically verified, demonstrating that the training strategy is able to dynamically recognize 3D objects in a relatively stable way.

## 1. Introduction

Deep learning has become a prevalent machine learning method [1,2,3] in recent years. As demonstrated, a multi-layer artificial neural network imitates the human learning process and then uncovers the hidden target patterns from a large amount of data, which can be comparable or even superior to the human recognition ability. With the rapid progress in deep learning, various electronic neural networks (ENNs) currently play a crucial role in solving target classification and recognition problems. The application of ENNs has extended into many fields, including computer vision [4,5,6,7,8,9] and neural language processing [10,11,12,13]. To address increasingly complicated problems, the fundamental architecture of ENNs is becoming accumulatively complex. However, there are still some flaws in conventional ENNs, such as their dramatic energy consumption and limited computational speed, which lead to the poor coupling of micro-sensors. At present, the scale of integrated chips is mesoscopic; thus, their development is seriously constrained by Moore’s law [14] and quantum tunneling. Therefore, it is crucial to find a way to address the problems above.

More recently, the diffractive deep neural network (D^2^NN), as a novel physical neural network with an all-optical deep learning framework, has been utilized to realize target classification and is practically constructed using several common methods, such as 3D-printing [15], femtosecond photon direct writing [16], and the wet etching technique. Due to the isomorphism between the forward propagation of dense neural networks and the Huygens–Fresnel principle, the D^2^NN can be implemented by sequential network layers connected by light diffraction to achieve the complete or local connection between neurons of each diffractive layer. This has the benefits of optical computing, such as theoretically operating at light speed with an extremely low consumption, and a parallel signal processing ability for optical computing [17,18,19,20]. The D^2^NN is evolving fast and is applied in many fields, such as Fourier-space D^2^NN [21], class-specific differential detection [22], multi-directional beam steering [23], and other applications [24,25,26,27,28,29,30]. Obviously, the D^2^NN has the potential to adaptively control the phase distribution of diffraction layers according to the complex amplitude distribution of diffractive beams to uncover the potential features of data and then learn the mapping relationship between the input light field and the actual label of the input object. Specifically, the D^2^NN has demonstrated excellent performance on many datasets, such as the Mixed National Institute of Standards and Technology database (MINIST) [15], the Fashion-MNIST dataset [21], and the CIFAR-10 dataset [22]. These studies confirm the possibility of using the optical neural network to achieve pattern recognition and conduct 2D object classification based on the D^2^NNs.

In the latest research, the optical neural network employing different coding methods can be combined with ensemble learning, and it enhances statistical inference and generalization capabilities [31]. Furthermore, the D^2^NN can take advantage of ensemble learning to learn light field patterns or images from multiple viewing simultaneously to recognize a 3D object. As shown, a kind of multiple-viewing D^2^NNs array (MDA) proposed by our research group can be efficiently used to achieve 3D object recognition under turbulence disturbance based on the Model-net dataset [32]. Therefore, the 3D targets can still be successfully recognized by the MDA even when the light field patterns or images based on multiple viewing are lost.

However, the performance of the MDA is poor when the object is translated, scaled, or shifted. The D^2^NN was demonstrated to be invariant to the shifting, scaling, or rotating variance of 2D objects [33]. However, unfortunately, it does not pay attention to factors such as shifting, scaling, or rotating in actual 3D objects in the real world. Therefore, it may be vulnerable to perturbation in actual 3D object recognition due to its sensitivity to the spatial information of targets. To improve the D^2^NN’s applicability to the real 3D world in terms of conducting complex object recognition, these object transformations should be considered as they frequently occur in various classification and recognition problems.

## 2. Method

To address the aforementioned problem, two measures were adopted in this paper: (1) A more robust MDA (*mr*-MDA) is proposed based on the plain MDA to put the base learner (BL) array on a surface instead of in a single line. The *mr*-MDA is composed of BLs from multiple viewing, and the bagging algorithm is used to make the final prediction, which comprehensively considers decisions obtained by all BLs. Therefore, the prediction error rate of the *mr*-MDA is theoretically decreased. In other words, the *mr*-MDA can receive more detailed information about the detected object so that it can still uncover the hidden abstracted clues to achieve classification under random disturbance; (2) a new training strategy is utilized to formulate the target transformations through random variables and then they are introduced as a disturbance to train the phases of diffractive layers, which thus enhances its capability to adapt to the uncertain transformation of input objects. This method was frequently employed in early ENNs to improve the robustness of the model, e.g., data enhancement technology [34], and to improve model generalization [35,36].

## 3. Diffraction Theory

Analysis of information forward-propagation is critical in conventional ENNs. The massive neurons of each layer are connected by forward-propagation based on electron flow. In the D^2^NN, the forward-propagation is based on Huygens’ principle, which is shown in Figure 1. There are many ways to achieve information forward-propagation corresponding to a fully connective neural network, a convolutional neural network, and a recurrent neural network, etc. In the D^2^NN, the neurons of each layers are fully connected by light diffraction, as shown in Figure 2.

According to Rayleigh–Sommerfeld diffraction theory, each neuron in the diffractive layer is considered as the source of a secondary wave, and it follows the following optical model [15]:(1)$$w_{i}^{L}\left(x,y,z\right)=\frac{z-z_{i}}{r^{2}}\left(\frac{1}{2\pi r}+\frac{1}{j\lambda }\right)\exp\left(\frac{j2\pi r}{\lambda }\right),$$
where L represents the L-th layer of the network, i denotes the neuron located at (x_i_, y_i_, z_i_) in the diffractive layer, r^2^ = (x − x_i_)^2^ + (y − y_i_)^2^ + (z − z_i_)^2^ and j^2^ = −1, and λ is the light wavelength. The amplitude and relative phase of this secondary wave are determined by the input wave of neurons and its transmission coefficient t. Specifically, the transmission coefficient can be expressed as
(2)$$t_{i}^{L}\left(x,y,z\right)=\alpha_{i}^{L}\left(x,y,z\right)\exp\left(j\phi_{i}^{L-1}\left(x,y,z\right)\right),$$
where the α and ϕ represent the amplitude modulation and phase modulation of the input light wave by one neuron, respectively. In the design process of the phase-modulated diffractive layer, we can use isotropic materials to make it. By setting the height of each point on the surface, the phase modulation of every neuron can be controlled. Therefore, one can write the output function of the i-th neuron in L-th layer as follows:(3)$$n_{i}^{L}\left(x,y,z\right)=w_{i}^{L}\left(x,y,z\right)\cdot t_{i}^{L}\left(x,y,z\right)\cdot \underset{k}{\sum }n_{k}^{L-1}\left(x,y,z\right)$$

Considering the information propagation of monochromatic light from a 2D plane, the Raleigh–Sommerfeld diffraction solution can be expressed as a convolution integral [37]:(4)$$U_{L}\left(x,y\right)=\iint U_{L-1}\left(\xi ,\eta \right)\cdot h\left(x-\xi ,y-\eta \right)d\xi d\eta ,$$
(5)$$h\left(x,y\right)=\frac{z}{j\lambda }\cdot \frac{\exp\left(\mathrm{jkr}\right)}{r^{2}},$$
(6)$$h\left(x,y\right)=\frac{e^{\mathrm{jkz}}}{j\lambda z}\exp\left(\frac{\mathrm{jk}}{2z}\left(x^{2}+y^{2}\right)\right),$$
where U is the light field. The form of the Raleigh–Sommerfeld impulse response can be written as Equation (5) and k = 2π/λ. To facilitate the calculation, the Fresnel approximation is used; in this situation, the impulse response can be written as Equation (6). According to Equation (4) and the Fourier convolution theorem, it is equivalently written as
(7)$$U_{L}\left(x,y\right)={ℑ}^{-1}\left\{ℑ\left\{U_{L-1}\left(x,y\right)\right\}\cdot H\left(f_{x},f_{y}\right)\right\},$$
(8)$$H\left(f_{X},f_{Y}\right)=e^{\mathrm{jkz}}\exp\left(j\pi \lambda z\left(f_{X}^{2}+f_{Y}^{2}\right)\right),$$ where ℑ represents the fast Fourier transform, U_L_ is the plane wave in the L-th diffractive layer, ℑ^−1^ represents the reverse fast Fourier transform, and the H(f_x_,f_y_) is defined as the Raleigh–Sommerfeld transfer function given by Equation (8). The f_X_ and f_Y_ are independent frequency variables associated with x and y. Two input light fields are converted by the above transformation to obtain the corresponding diffractive light field in Figure 3.

## 4. Basic Architecture

The basic structure of the *mr*-MDA was remodified based on the original MDA and it can better adapt to the random shifting, scaling, or rotating transformations. Specifically, Figure 4 shows the configuration of the BLs in this paper, which was designed as a grid with two rows and three columns on the surface of a sphere. The red point represents a BL. The angle of adjacent BLs in the same row is demonstrated by *θ_v_*, and the angle difference of adjacent BLs in the same column is marked as *θ_h_*, which were designed to be at 30° and 15°, respectively. Compared with the original MDA, the additional BLs were placed in a distinct row, instead of all learners being in the same row, enabling them to receive more spatial information about the target in different viewings and then increase the independence between BLs. Taking advantage of this basic architecture, the hypothesis space of the *mr*-MDA is significantly extended, thus making it possible to adapt to targets with random disturbance and increasing the probability of finding an optimal answer.

The detailed inference process for one sample of the *mr*-MDA is demonstrated by the relationships in
(9)$$w=\frac{\sum_{k=1}^{\mathrm{NS}}I\left[y_{k}=F_{i,j}\left(x_{k}\right)\right]}{\mathrm{NS}},$$
(10)$$s=\sum_{i=1}^{m}\sum_{j=1}^{n}w_{i,j}\cdot \alpha_{i,j},$$
(11)$$\mathrm{pr}=\underset{0<=i<M}{\mathrm{argmax}}\left(s_{i}\right),$$

For a more convenient explanation, the subscripts of i and j demonstrate the BL located in the ith row and the jth column in the *mr*-MDA. In Equation (9), w_i,j_ ∈R denotes the weight of the BLs, NS denotes the number of samples in the validation dataset, and I[∙] represents the indicator function, which is defined as 1 or 0, respectively, when the condition in the brackets is true or false. In addition, x_k_ is the kth sample in the validation, and its true label is marked as y_k_. F_i,j_(∙), demonstrating a mapping function of the corresponding BLs, which propagates the light field information of the input image through several diffractive layers to consequently shape a specific spot image, leading to the prediction of x_k_. The w_i,j_ is obtained based on the performance of the BLs during the validation stage and then represents the reliability of one BL; its inference can be described as shown in Equation (9). Consequently, the prediction of the *mr*-MDA is mainly considered by the most dependable BL, and thus is affected by the rest of the BLs. The M indicates the number of categories in a dataset, which is set as 10. The α_i, j_ ∈R^1^^×M^ denotes a vector of the prediction result of the BL for one sample, and its ith column value represents the correct rate at which the target is classified as the ith category. The s = (s^(0)^, s^(1)^, … s^(M−1)^)∈R^1^^×M^ is an integrated prediction output vector (IPOV) of the *mr*-MDA, and the significance of each column value is the same as each column of α_i, j_. In Equation (11), the pr denotes the prediction result of the *mr*-MDA, which is the column number of the highest value in the IPOV.

In summary, Equation (10) guides the concrete realization of the weighted voting algorithm, and the *mr*-MDA draws a conclusion based on the collective wisdom, which helps the *mr*-MDA to accurately achieve classification despite the random disturbance already introduced into a 3D target. Additionally, Equation (10) conveys a simple linear operation, and its time complexity is only determined by the scale of the BL array, such as approximately O(*m* × *n*), which can be negligible during an entire calculation. Hence, it will exhibit an inappreciable impact on the costing time of the entire optical network system, which enables the entire system to perform an ultra-high-speed operation despite the participation of electronic devices.

As shown in Figure 5, a multiple BL of the *mr*-MDA will accept light fields from the different viewing angles of a 3D target. These light fields present more information, such as the depth of the target, than that of a single light field, which is the principal reason as to why the *mr*-MDA achieves higher accuracy even when the 3D target randomly occurs based on a shifting, rotating or scaling operation. The predicted light spot distributions are obtained by all BLs and then are transmitted to the computer or electronic circuit. Furthermore, all predicted light fields are integrated as an ensemble light field to obtain the IPOV according to the weighted voting method: an implementation of the bagging algorithm. To facilitate the description, each light spot of the ensemble light field is numbered as shown in this picture, which represents a specific category of the target. Finally, every column value of the IPOV represents the confidence level to which the target belongs in the corresponding category. The category with the highest value is the final prediction result of the *mr*-MDA, which is demonstrated by Equation (11). The multiple BLs are placed in different positions on the sphere with a customized distribution in order to receive the required light field images from different viewing angles. Determinations will be drawn by all BLs, which are integrated into an ensemble light field, and the final prediction is inferred by the *mr*-MDA by choosing the brightest light spot.

As shown in Figure 6, an essential BL of the *mr*-MDA is designed as a series of diffractive layers and some existing diffractive residual blocks. Figure 6a demonstrates the key structure of a BL. In our work, the distance between each diffractive layer and between the last diffractive layer and the output layer was set to 22.72λ and 68.18λ, respectively, where λ denotes the light wavelength. Every network layer of the BL was set in a 72.72λ × 72.72λ format, with 200 × 200 layered neurons in a light field scenario and a 2.2 μm wavelength so it can be integrated into micro-detectors. There are three diffractive layers in the front of residual blocks, two diffractive layers in the back, and a photodetector as an output layer. For enhancing the capacity of the *mr*-MDA, there are two residual blocks and each of them is made of three diffractive layers. The BLs of the *mr*-MDA were trained by calculating the cross-entropy loss of the forward propagation, which was simulated. Each layer can be fabricated as a physical entity. The output of each BL is transmitted to the computer or electronic circuit to participate in weighted voting to form a final output as a prediction.

For a given 3D object, every trained BL can be used to obtain a customized optical spot pattern based on a specific single viewing light field. All BLs adopt cross entropy as a loss function for calculating the output loss and further employ the Adam algorithm and error backpropagation to update the phase of the diffractive layer. The loss function for one training sample is demonstrated by the expressions in
(12)$$\alpha =\frac{\exp(I)}{\sum_{i=0}^{M-1}\exp(I^{\left(i\right)})},$$
(13)$$L=-\frac{1}{M}\sum_{i=0}^{M-1}\beta^{\left(i\right)}{\log\alpha }^{\left(i\right)}+\left(1-\beta^{\left(i\right)}\right)\log(1-\alpha^{\left(i\right)}),$$
where I = (I^(0)^, I^(1)^, … I^(M−1)^)∈R^1^^×M^ denotes a light intensity vector of the spot in the output layer of the BL, and I^(i)^ is the light intensity of the ith light spot. In Equation (13), L is the cross entropy, which statistically indicates the difference between the prediction of the *mr*-MDA and the true target label. The α, which is one of α_i,j_ and a normalized vector of the light intensity vector, can be calculated using a soft-max function, as indicated in Equation (12). The β^(i)^ denotes the i^th^ column of the true one-hot label, and its value can be selected as 1 or 0.

According to previous studies, the original deep neural network has several defects, such as a large number of model parameters, a long training time, and the risk of overfitting. As demonstrated, the residual deep neural network [38] can be utilized to help the network to overcome the above shortcomings in the case of the same layer number. Therefore, it is possible to build a deeper ENN that can better uncover potential patterns or images from more complex samples. It was verified that the scheme is realizable in an optical neural network and has excellent performance compared with the architecture of the plain D^2^NN in a recent study [39]. Consequently, to enhance the expression capability of the *mr*-MDA to cope with shifting, scaling, or rotating targets and to further avoid the problem of vanishing gradients and overfitting in model training, each unit of the BLs should contain diffractive residual blocks. Figure 6b shows the fundamental construction of the diffractive residual block for base learning. Light splitting and focusing can be achieved in the residual block by employing a beam splitter and reflecting mirror.

## 5. Experimental Preparation 

To better explain the transformations mentioned, some imaging results received by the BLs located in distinct perspectives after introducing random disturbance into one of samples from the dataset are demonstrated in Figure 7. The proposed training strategy introduces random shifting, scaling, and rotating variance into the original targets, and the degree of random object transformation is respectively represented by independently and uniformly distributed random variables. The parameters of Δ, *K*, and Θ are all three-dimensional vectors used to describe spatial information after undergoing some uncertain beam transformations. Δ *=* (Δ*_x_*, Δ*_y_*, Δ*_z_*) indicates that the input object presents some shifts of Δ*_x_*, Δ*_y_*, and Δ*_z_* in the *x*, *y,* and *z* directions, respectively. Each of their column values is defined as a uniformly distributed variable, for instance, Δ*_x_* ~ *U* (−*δ*, *δ*), Δ*_y_* ~ *U* (−*δ*, *δ*), Δ*_z_* ~ *U* (−*δ*, *δ*). The hyperparameter *δ* represents the displacement level of the target, and *δ* ∈ [0, 0.7] in this letter. Therefore, the increase in the value of *δ* will result in an increased possibility and a variance range of the target displacement, which will also introduce a greater disturbance into the model. To appropriately quantify the influence of Δ on the image formed by a single viewing, the significance of Δ*_x_*, Δ*_y_*, and Δ*_z_* is carefully described as shown in Figure 8. Both the scaling and rotating parameters are individually defined as *K=* (*k_x_*, *k_y_*, *k_z_*) and Θ*=* (Θ*_x_*, Θ*_y_*, Θ*_z_*), which means that the original targets are scaled and rotated along the *x* and *y* and *z* directions, respectively. In order to maintain the proper object proportion, the *K* is further designed into *k_x_ = k_y_ = k_z_~U* (1 − *ε*, 1 + *ε*) and *ε* ∈ [0, 0.7] in this letter. To this end, each column of Θ individually conforms to a uniform distribution of *U* (*−θ*, *θ*) and *θ* ∈ [0°, 60°] in this manuscript. For the sake of convenience, a parameter of *view_i,j_(v_i,j_)* was used to represent the BL located in the *i*th row and the *j*th column in our BLs’ array, and they will receive the input light field image of the target in different views.

For the given hyperparameters of *δ*, *ε*, and *θ*, the range of a random disturbance introduced into the original target is controllable. Therefore, we can adjust their values to illustrate the impact of the object’s shifting, scaling, and rotating variance on the prediction results of the *mr*-MDA mode. Furthermore, by assigning different values to the variables on both the training dataset and the blind test data, we demonstrate the prospective performance of the *mr*-MDA using a new training strategy.

## 6. Results and Discussion

The performance comparison of the *mr*-MDA, the BLs, and the MDA is demonstrated in Figure 9a–c. It was verified that the architecture of the *mr*-MDA is more capable of adapting the 3D target, which already has random disturbance introduced, than the MDA. This is significant progress compared with a single viewing BL. Specifically, the classification accuracy of the *mr*-MDA at all sampling points is superior as compared to the other two models mentioned. According to the experiments, it can be seen that the test accuracy of the *mr*-MDA is higher than that of the BL by ~30% and the MDA by ~15% under the situation of target shifting, by ~30% and ~10% under target scaling, and by ~15% and ~10% under target rotating. The results mentioned above were obtained based on strong interference (more details are shown in Figure 9). In addition, as the values of the variables increase, the recognition capabilities of the single viewing BL, MDA, and *mr*-MDA decline significantly. However, the performance of the *mr*-MDA declines more slowly owing to the *mr*-MDA receiving more target spatial information. It should be noted that the *mr*-MDA demonstrates a larger hypothesis space in statistics, which is sufficient to uncover more complex patterns. Moreover, concrete data as regards specific interference are shown in Table 1. The decision making combined with multiple perspectives is more accurate than only relying on one single perspective, and this phenomenon becomes more obvious with the increase in the number of perspectives when interference is introduced. However, this difference is not apparent with no interference.

The phase diagram in Figure 9d–g demonstrates that after introducing disturbance into the training dataset, the phase of the corresponding diffractive layer also changed significantly compared with that of the standard phase. Specifically, the spiral phase shown in Figure 9g is more evident and adjacent to the center of the figure. In contrast, the other spiral phases in Figure 9d–f are distinct, owing to the diffraction layer of the *mr*-MDA taking random disturbance into account in the error propagation of the training phase and adaptively resisting it. Furthermore, the *mr*-MDA is driven based on ensemble learning in the D^2^NN, so its performance is closely related to the independence between the BLs, which leads to a superior performance as compared to the plain D^2^NN and MDA as random disturbance increases.

The detailed inference of the *mr*-MDA is the same as that of the MDA; we present two of the predictions in Figure 10. Every trained BL individually outputs a specific spot pattern, and some are in line with the expectations, but others are wrong. According to the BL weight, the *mr*-MDA draws a conclusion based on collective wisdom using a weighted voting algorithm. The brightest light spot exists at the desired position (as shown by the white arrow) in the subfigures with a red frame, which indicates a correct prediction result. In the sample on the left in Figure 10, the *mr*-MDA achieves a correct prediction despite two BLs already making the wrong decision, because each BL is given the appropriate weight during training; thus, it makes the correct decision even when more than half of the BLs make wrong predictions, as shown on the right in Figure 10. In addition, Figure 10 also shows that the light distribution of the BL is significantly different from the light distribution of the *mr*-MDA, and the latter is only prominent at one certain point, which does not interfere with the final decision.

As demonstrated, each BL outputs a specific spot pattern after receiving light field images from different viewing angles, and then the *mr*-MDA determines the target’s category based on the BLs’ weighted voting (e.g., the prediction accuracy of each BL on the validation dataset = [0.471, 0.357, 0.401, 0.458, 0.411, 0.454] in Figure 10, so the corresponding BL weight = [0.185, 0.140, 0.157, 0.179, 0.161,0.178]). In the end, the final spot pattern is obtained by weighted voting, which has the highest light intensity at the correct spot position. The weighted voting algorithm allows the *mr*-MDA to consider all results of BLs and then adaptively favor the more reliable ones, which explains why the BL making a wrong decision has a smaller weight but the final prediction is correct, as shown in Figure 10. This pattern of decision making not only improves the fault tolerance rate of the whole micro-system, but also enables the whole optical process to work in parallel.

Apparently, the modified architecture of the *mr*-MDA is more robust than that of the MDA. To further improve the robustness of the *mr*-MDA, we adopted a training strategy that defines both parameters of *δ*_train_ and *δ*_test_ as the shifting variable in the training dataset and test dataset, both *ε*_train_ and ε_test_ as the scaling variable in the training dataset and test dataset, and both *θ*_train_ and *θ*_test_ as the rotating variable in the training dataset and test dataset. The *mr*-MDA has prior knowledge and is not sensitive to spatial information concerning shifting, scaling, or rotating while achieving 3D target classification. Figure 11 shows the excellent results obtained by employing the new training strategy.

The *mr*-MDA trained by the shifting, scaling, and rotating datasets presents several obvious advantages compared with that trained only by the standard dataset. As the disturbance on the test dataset increases, the performance of the *mr*-MDA using the normal training strategy declines sharply, while the performances after employing the new training strategy drop slowly or remain almost stable. Specifically, Figure 11a shows that a standard *mr*-MDA can achieve excellent blind test accuracy in a weak disturbance. However, it also obtains a small mean and high variance behavior over the entire experiment (e.g., the standard *mr*-MDA achieved a ~85.9% test accuracy when *δ*_test_ = 0, *ε*_test_ = 0, and *θ*_test_ = 0, but its final average test accuracy was only ~43.1%, ~71.2%, and ~58.0%, and the variance was ~5.7%, ~1.3%, and ~3.9%), which is not enough to conduct satisfactory 3D target classification after random disturbance has already been introduced. The *mr*-MDA trained with the standard dataset lacks prior knowledge of the 3D spatial information of targets and only learns the standard spatial information, such as the position, size, and direction of targets. In statistics, it is impossible to find an optimal solution to adapt to the uncertain object. Therefore, when interference is introduced into the training data, the mean and variance of the *mr*-MDA over the entire experiment are greatly improved (e.g., when *δ*_train_ = 0.4, the average accuracy reached ~57% and its variance was only ~0.7%. When *ε*_train_ = 0.5, the average accuracy reached ~78.5% and its variance was ~ 0.2%. When *θ*_train_ = 30, the average accuracy reached ~65.9% and its variance was 1.1%). Remarkably, excessive interference in the training dataset can reduce the prediction accuracy, but the variance still decreases (e.g., as shown in Figure 11, when *δ*_train_ = 0.7, *ε*_train_ = 0.7, and *θ*_train_ = 60, the variance was approximately zero but their averages were ~54.5% and ~77.2% and ~56.5%).

In summary, the proposed *mr*-MDA can effectively recognize 3D objects that are shifted, scaled, and rotated as a result of its basic architecture, which already enhances the independence between the BLs. Furthermore, it was verified that the proposed training strategy can appropriately increase the mean value of the final decision accuracy and decrease its variance, which plays an essential role in using the D^2^NN to stably recognize 3D objects. In this work, the *mr*-MDA was trained based on the Model-net dataset using Python 3.8.9 and the framework of Tensorflow 2.0, and we chose blender 2.79b to obtain a multi-viewing imaging light field of 3D targets. The training and testing of the *mr*-MDA were run on a PC with Intel Core i7-10700 CPU (2.90GHZ) and the GeForce RTX 2070 (NVIDIA). It generally took ~20 min to train one BL and thus cost ~375.12KB in memory consumption. Since the BLs were trained independently, each BL was trained using different distributed server nodes.

## 7. Conclusions

According to this experiment, it was concluded that the D^2^NN can indeed theoretically accomplish various complicated tasks. The original MDA model was remarkably improved and a new training strategy utilized to obviously expand the applications of the *mr*-MDA in terms of accomplishing more difficult classification. The *mr*-MDA model dramatically enhances the independence between the BLs and it exhibits superior performance in 3D object classification under random interference compared with the MDA and the plain D^2^NN. In addition, the *mr*-MDA, employing the proposed training strategy, is insensitive to the spatial information of targets, which is a key feature when utilizing an arrayed photonic integrated device to recognize complex 3D objects in a dynamic and rapid manner with low-energy consumption. Although it is currently difficult to perfectly align more diffractive layers, related work has been published that analyzes the errors in model placement [40]. In addition, the model used in this article provides a way in which to avoid excessively deep network layers. More perspectives can be used to replace the deeper network and couple it to the front of the sensor, because optical computing is parallel and high-speed, and the system size is micro-scale.

## Figures and Tables

**Figure 1 sensors-22-07754-f001:**
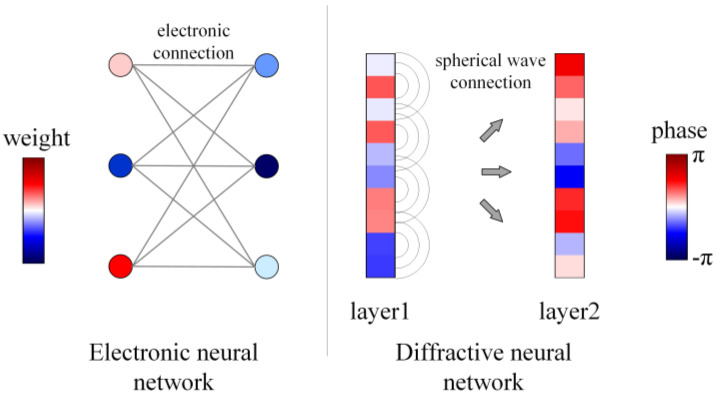
The comparation of two neural networks.

**Figure 2 sensors-22-07754-f002:**
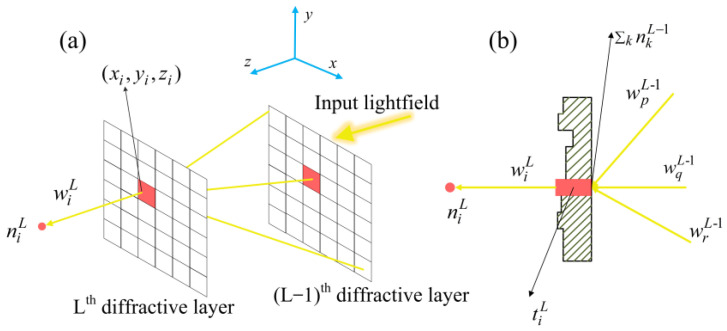
The illustration of forward-propagation in the D^2^NN. (**a**) General schematic of forward-propagation. (**b**) Cross-section of a neuron in the diffractive layer.

**Figure 3 sensors-22-07754-f003:**
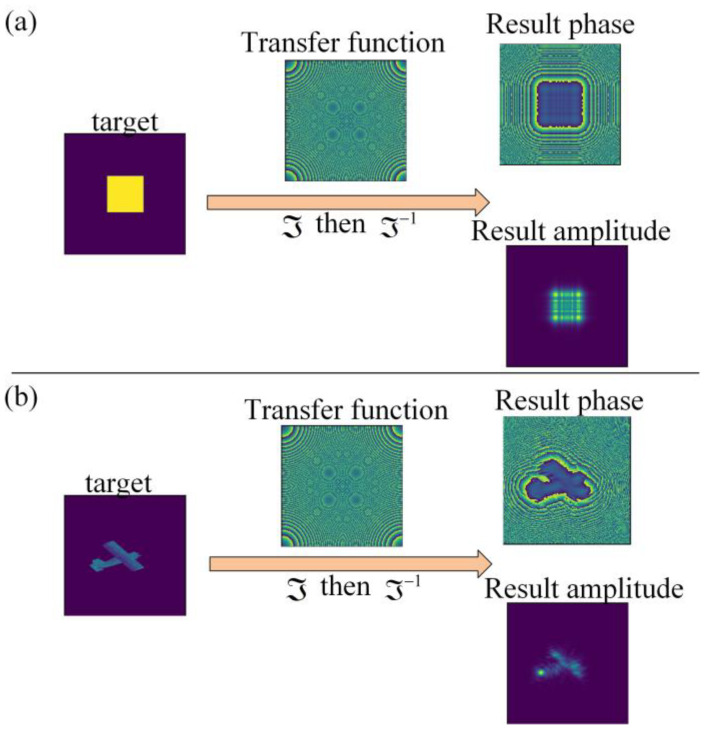
Diffraction simulation display. (**a**) Diffraction of square. (**b**) Diffraction of real target from dataset.

**Figure 4 sensors-22-07754-f004:**
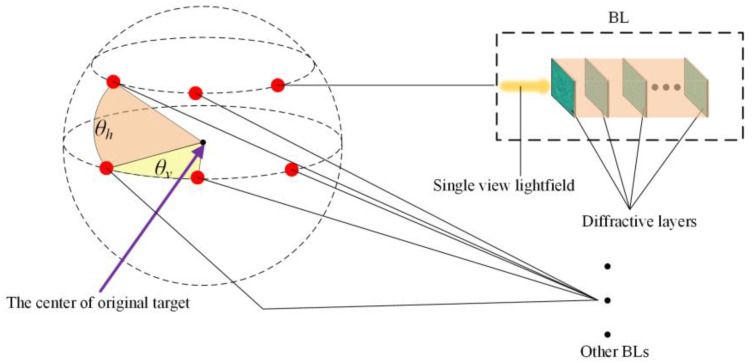
Typical distribution character of the BLs.

**Figure 5 sensors-22-07754-f005:**
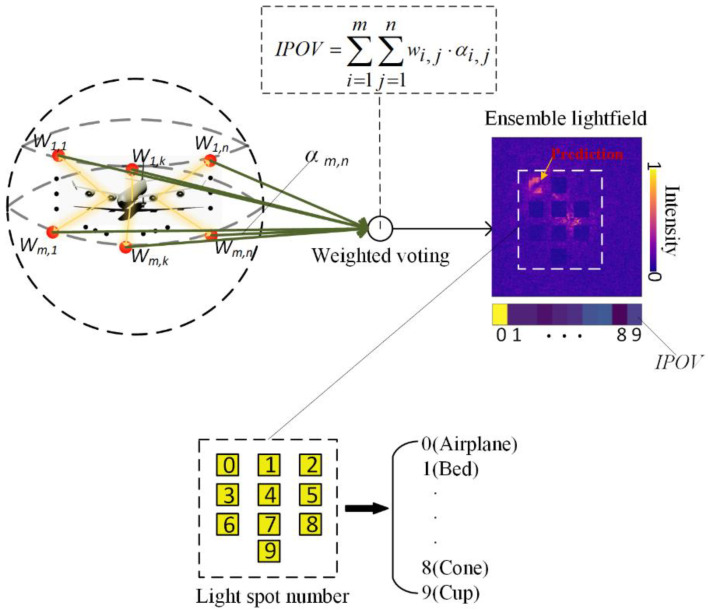
Schematic diagram of the *mr*-MDA.

**Figure 6 sensors-22-07754-f006:**
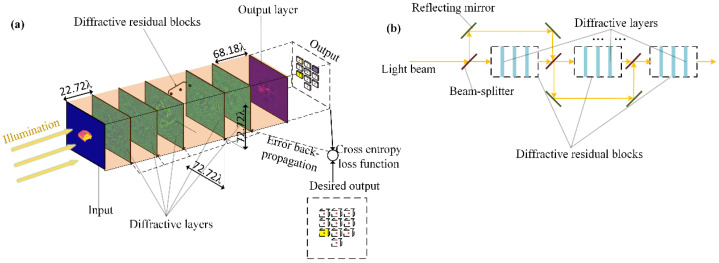
An essential BL of the *mr*-MDA was designed as a series of diffractive layers with some diffractive blocks. (**a**) Basic architecture of an essential BL. (**b**) Core architecture of the diffractive residual blocks in the BL.

**Figure 7 sensors-22-07754-f007:**
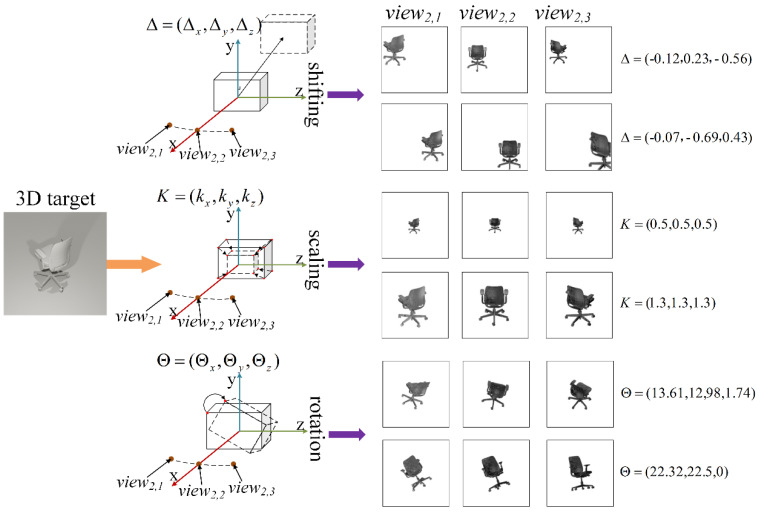
Illustration of target transformation.

**Figure 8 sensors-22-07754-f008:**
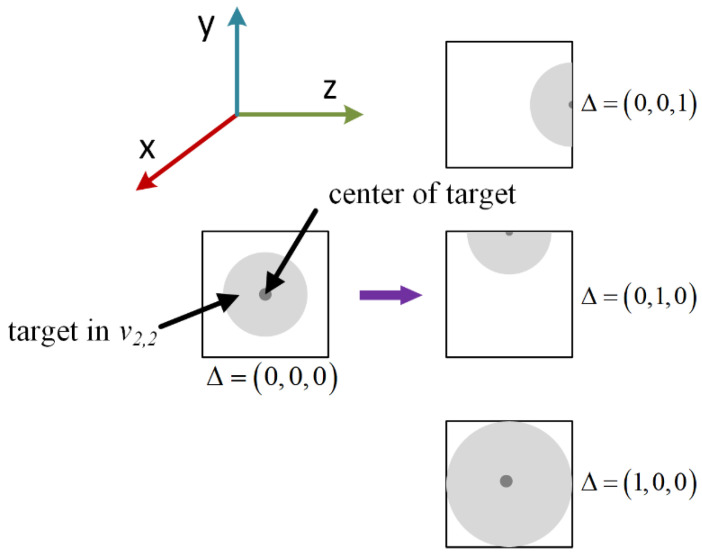
Definition of the parameter Δ.

**Figure 9 sensors-22-07754-f009:**
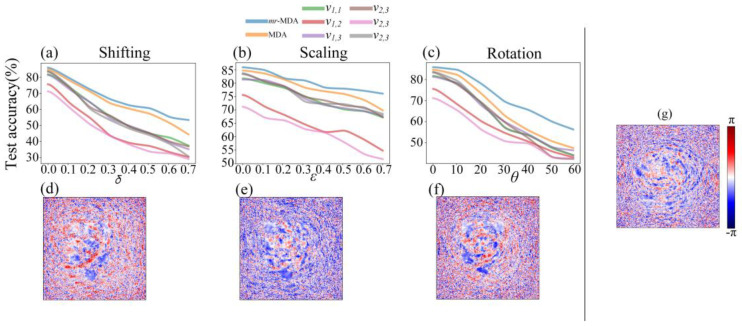
Performance with different random disturbances corresponding to the *mr*-MDA, the BLs, and the plain MDA. The subfigures (**a**–**c**) respectively represent the curves of the test accuracy under shifting, scaling, and rotating disturbance. The subgraphs (**d**–**g**) respectively denote the phase image of one diffractive layer of the *mr*-MDA trained by shifting (δ = 0.6), scaling (ε = 0.8), and rotating (θ = 50°), and the standard (δ = 0, ε = 0, θ = 0) training dataset.

**Figure 10 sensors-22-07754-f010:**
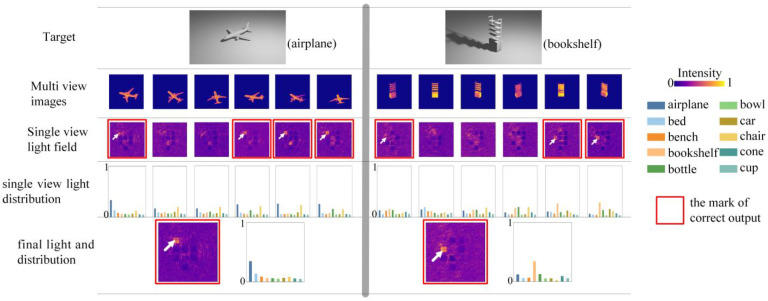
Detailed inference of the *mr*-MDA.

**Figure 11 sensors-22-07754-f011:**
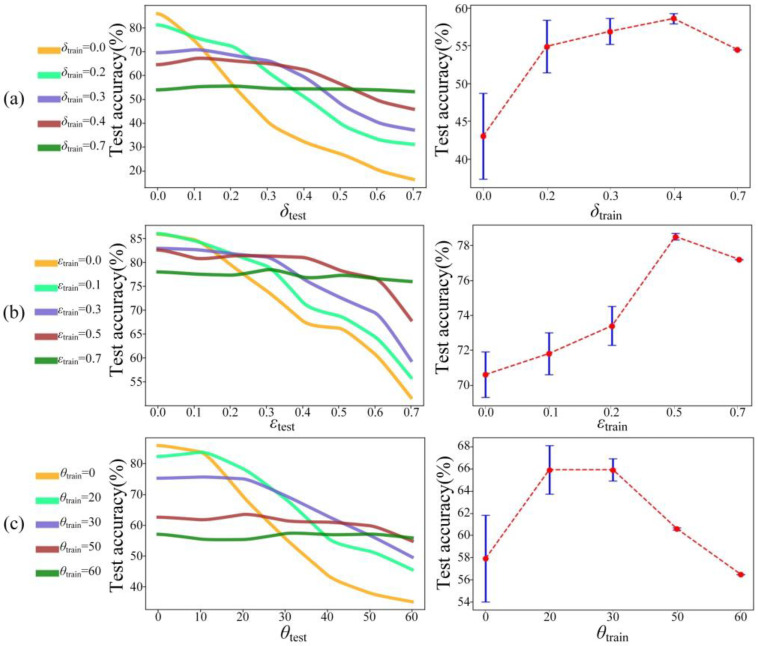
Comparison of the performances using different training strategies. The comparison between the inference accuracies of the *mr*-MDA resisting target shifting (left) (**a**), resisting target scaling (left) (**b**), and resisting target rotation (left) (**c**). The variance of each curve is in the left picture (right). The value corresponding to the red point in the right subgraphs represents the mean, and the distance from the red point to the upper and lower ends denotes the variance.

**Table 1 sensors-22-07754-t001:** The accuracy of different models under specific circumstances.

Model	Not Inference (%)	*δ* = 0.6 (%)	*ε* = 0.8 (%)	*θ* = 50° (%)
single BL	83.1	41.2	72.3	49.2
3-MDA	84.3	49.3	74.8	52.7
6-MDA	87.2	58.3	77.2	63.1

## Data Availability

The data supporting reported results can be found at http://modelnet.cs.princeton.edu/.

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
