# Peer review of "The mr-MDA: An Invariant to Shifting, Scaling, and Rotating Variance for 3D Object Recognition Using Diffractive Deep Neural Network"

_sensors, 2022, doi:10.3390/s22207754_

Round 1
Reviewer 1 Report
Comments to the Author
In this work, based on the multiple-view D2NN array (MDA), 3D objects can be recognized in a high-speed and dynamic way. Through numerical verification, the author finds that it is invariant to typical shifting, scaling and rotating variance of the target in complicated situation. This is a very interesting work. The paper is well organized and clearly written, which can be accepted for publication, provided that the authors further improve the work by addressing the comments as follows.
Minor comments,
1. The network proposed by the author is very complex, but the accuracy is still very low (no more than 90%) when shifting, scaling and rotating variance for 3D objects are not considered. Whether the author can analyze the causes of low accuracy and the solutions.
2. For the optical neural network with diffractive residual blocks, what is its diffraction efficiency theoretically?
3. In Figure 9, it seems that the neural network proposed by the author has no obvious improvement in shifting, scaling and rotating object recognition compared with the previous MDA. What is the main reason and how to improve it?
4. What are the advantages of the neural network proposed by the author over previous methods for 2D object recognition?
5. In the introduction, the author mentioned the extensive application of electronic neural networks (ENNs) and the diffractive deep neural network (D2NN). The description is too simple. More previous works are suggested to be mentioned (Gu, Min, et al. "Optically digitalized holography: a perspective for all-optical machine learning." Engineering 5.3 (2019): 363-365. Luan, Haitao, et al. "768-ary Laguerre-Gaussian-mode shift keying free-space optical communication based on convolutional neural networks." Optics Express 29.13 (2021): 19807-19818.) to be helpful to wide readers.
Reviewer 2 Report
lines 44:48 please explain what architecture you mean. Do that statement include ReLU activation function? How regularization (BathNorm or other) can be included into this area.
lines 52:59 add link,
all equations: please, add conventional denotation for all variables you use.
section 4: i do not find any explanation of non-linearity (activation function) beside eq 11 (last layer).
please, do not use such long sentences, it leads to complexity of reading.
